# Isometric Propagation Network for Generalized Zero-shot Learning

**Lu Liu[†], Tianyi Zhou[‡], Guodong Long[†], Jing Jiang[†], Xuanyi Dong[†], Chengqi Zhang[†]**
† University of Technology Sydney, ‡ University of Washington
Corresponding to: lu.liu.cs@icloud.com

## Abstract

Zero-shot learning (ZSL) aims to classify images of an unseen class only based on a few attributes describing that class but no access to any training sample. A popular strategy is to learn a mapping between the semantic space of class attributes and the visual space of images based on the seen classes and their data. Thus, an unseen class image can be ideally mapped to its corresponding class attributes. The key challenge is how to align the representations in the two spaces. For most ZSL settings, the attributes for each seen/unseen class are only represented by a vector while the seen-class data provide much more information. Thus, the imbalanced supervision from the semantic and the visual space can make the learned mapping easily overfitting to the seen classes. To resolve this problem, we propose Isometric Propagation Network (IPN), which learns to strengthen the relation between classes within each space and align the class dependency in the two spaces. Specifically, IPN learns to propagate the class representations on an auto-generated graph within each space. In contrast to only aligning the resulted static representation, we regularize the two *dynamic* propagation *procedures* to be isometric in terms of the two graphs' edge weights per step by minimizing a consistency loss between them. IPN achieves state-of-the-art performance on three popular ZSL benchmarks. To evaluate the generalization capability of IPN, we further build two larger benchmarks with more diverse unseen classes, and demonstrate the advantages of IPN on them.

## 1 Introduction

One primary challenge on the track from artificial intelligence to human-level intelligence is to improve the generalization capability of machine learning models to unseen problems. While most supervised learning methods focus on generalization to unseen data from training task/classes, zero-shot learning (ZSL) (Larochelle et al., 2008; Lampert et al., 2014; Xian et al., 2019a) has a more ambitious goal targeting the generalization to new tasks and unseen classes. In the context of image classification, given images from some training classes, ZSL aims to classify new images from unseen classes with zero training image available. Without training data, it is impossible to directly learn a mapping from the input space to the unseen classes. Hence, recent works introduced learning to map between the semantic space and visual space (Zhu et al., 2019a; Li et al., 2017; Jiang et al., 2018) so that the query image representation and the class representation can be mapped to a shared space for comparison.

However, learning to align the representation usually leads to overfitting on seen classes (Liu et al., 2018). One of the reasons is that in most zero-shot learning settings, the number of images per class is hundreds while the semantic information provided for one class is only limited to one vector. Thus, the mapping function is easily overfitting to the seen classes when trained using the small set of attributes. The learned model typically has imbalanced performance on samples from the seen classes and unseen classes, i.e., strong when predicting samples from the seen classes but struggles in the prediction for samples in the unseen classes.

In this paper, we propose "*Isometric Propagation Network* (IPN)" which learns to dynamically interact between the visual space and the semantic space. Within each space, IPN uses an attention module to generate a category graph, and then perform multiple steps of propagation on the graph.

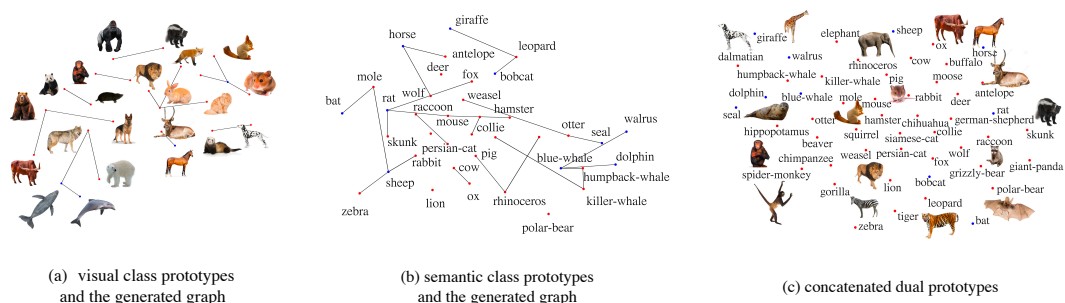

(a) visual class prototypes
and the generated graph

(b) semantic class prototypes
and the generated graph

(c) concatenated dual prototypes

Figure 1: t-SNE (Maaten & Hinton, 2008) of the prototypes produced by IPN on AWA2. Blue/red points represent unseen/seen classes.

In every step, the distribution of the attention scores generated in each space are regularized to be isometric so that implicitly aligns the relationships between classes in two spaces. Due to different motifs of the generated graphs, the regularizer for the isometry between the two distributions are provided with sufficient training supervisions and can potentially generalize better on unseen classes. To get more diverse graph motifs, we also apply episodic training which samples different subset of classes as the nodes for every training episode rather than the whole set of seen classes all the time.

To evaluate IPN, we compared it with several state-of-the-art ZSL methods on three popular ZSL benchmarks, i.e., AWA2 (Xian et al., 2019a), CUB (Welinder et al., 2010) and aPY (Farhadi et al., 2009). To test the generalization ability on more diverse unseen classes rather than similar classes to the seen classes, e.g., all animals for AWA2 and all birds for CUB, we also evaluate IPN on two new large-scale datasets extracted from *tiered*ImageNet (Ren et al., 2018), i.e., *tiered*ImageNet-Mixed and *tiered*ImageNet-Segregated. IPN consistently achieves state-of-the-art performance in terms of the harmonic mean of the accuracy on unseen and seen classes. We show the ablation study of each component in our model and visualize the final learned representation in each space in Figure 1. It shows the class representations in each space learned by IPN are evenly distributed and connected smoothly between the seen and unseen classes. The two spaces can also reflect each other.

## 2 RELATED WORKS

Most ZSL methods can be categorized as either discriminative or generative. Discriminative methods generate the classifiers for unseen classes from its attributes by learning a mapping from a input image's visual features to the associated class's attributes on training data of seen classes (Frome et al., 2013; Akata et al., 2015a;b; Romera-Paredes & Torr, 2015; Kodirov et al., 2017; Socher et al., 2013; Cacheux et al., 2019; Li et al., 2019b), or by relating similar classes according to their shared properties, e.g., semantic features (Norouzi et al., 2013; Chi et al., 2021) or phantom classes (Changpinyo et al., 2016), etc. In contrast, generative methods employ the recently popular image generation models (Goodfellow et al., 2014) to synthesize images of unseen classes, which reduces ZSL to supervised learning (Huang et al., 2019) but usually increases the training cost and model complexity. In addition, the study of unsupervised learning (Fan et al., 2021; Liu et al., 2021), meta learning (Santoro et al., 2016; Liu et al., 2019b;a; Ding et al., 2020; 2021), or architecture design (Dong et al., 2021; Dong & Yang, 2019) can also be used improve the representation ability of visual or semantic features.

**Alignment between two spaces in ZSL** Frome et al. (2013); Socher et al. (2013); Zhang et al. (2017) proposed to transform representations either to the visual space or the semantic space for distance for comparison. Li et al. (2017) introduces how to optimize the semantic space so that the two spaces can be better aligned. Jiang et al. (2018); Liu et al. (2020) tries to map the class representations from two spaces to another common space. Comparatively, our IPN does not align two spaces by feature transformation but by regularizing the isometry of the intermediate output from two propagation model.

**Graph in ZSL** Graph structures have already been explored for their benefits to some discriminative ZSL models, e.g., Wang et al. (2018); Kampffmeyer et al. (2019) learn to generate the weights of a fully-connected classifier for a given class based on its semantic features computed on a pre-

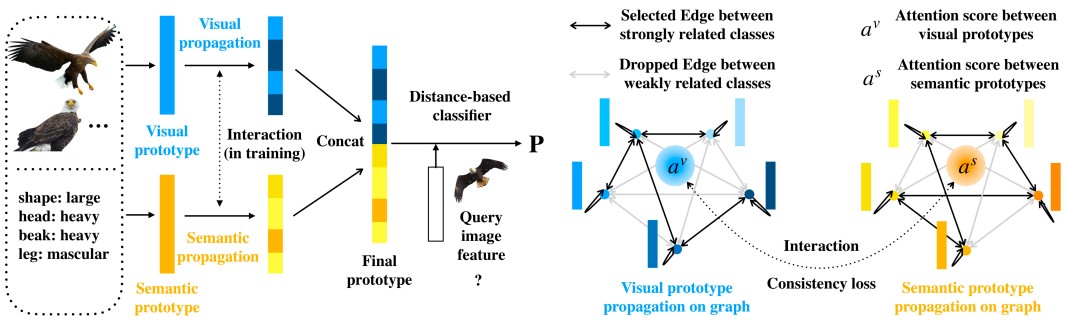

Figure 2: **LEFT: The pipeline of IPN.** First, we initialize the visual and semantic prototypes from seen classes's data and class attributes. Then, prototype propagation in each space can be performed over multiple steps. Finally, the concatenated prototypes form the support set of a distance based classifier. **RIGHT: Isometric propagation.** we propagate prototypes of each class to other connected classes on the graph using attention module. The attention score is regularized to be isometric.

defined semantic graph using a graph convolutional networks (GCN). They require (1) a pre-trained CNN whose last fully-connected layer provides the ground truth of the classifiers ZSL aims to generate; and (2) an *existing* knowledge graph as the input of GCN. By contrast, our method learns to automatically generate the graph and the class prototypes composing the support set of a distance-based classifier. Hence, our method does not require a pre-trained CNN to provide a ground truth for the classifier. In addition, we train a modified graph attention network (GAN) (rather than a GCN) to generate the prototypes, so it learns both the class adjacency and edges by itself and can provide more accurate modeling of class dependency in scenarios when no knowledge graph is provided. Tong et al. (2019) and Xie et al. (2020) also leverage the idea of graph to transform the feature extracted from the CNN while our graph serves as the paths for propagation. It requires an off-the-shelf graph generated by ConceptNet (Speer et al., 2016), while IPN applies prototype propagation and learns to construct the graph in an end-to-end manner.

**Attention in ZSL** The attention module has been applied for zero-shot learning for localization. Liu et al. (2019c) introduces localization on the semantic information and Xie et al. (2019) proposes localization on the visual feature. Zhu et al. (2019b) further extends it to multi-localization guided by semantic. Different from them, our attention serves as generating the weights for propagation.

Our episodic training strategy is inspired by the meta-learning method from (Santoro et al., 2016). Similar methods have been broadly studied in meta-learning and few-shot learning approaches (Finn et al., 2017; Snell et al., 2017; Sung et al., 2018) but rarely studied for ZSL in previous works. In this paper, we show that training ZSL model as a meta-learner can improve the generalization performance and mitigate overfitting.

## 3 PROBLEM SETTING

We consider the following setting of ZSL. The training set is composed of data-label pairs $(\boldsymbol{x}, y)$, where $\boldsymbol{x} \in \mathcal{X}$ is a sample from class $y$, and $y$ must be one of the seen classes in $\mathcal{Y}^{seen}$. During the test stage, given a test sample $\boldsymbol{x}$ of class $y$, the ZSL model achieved in training is expected to produce the correct prediction of $y$. In order to relate the unseen classes in test to the seen classes in training, we are also given the semantic embedding $\boldsymbol{s}_y$ of each class $y \in \mathcal{Y}^{seen} \cup \mathcal{Y}^{unseen}$. We further assume that seen and unseen classes are disjoint, i.e., $\mathcal{Y}^{seen} \cap \mathcal{Y}^{unseen} = \emptyset$. The test samples come from both $\mathcal{Y}^{seen}$ and $\mathcal{Y}^{unseen}$ in generalized zero-shot learning.

## 4 ISOMETRIC PROPAGATION NETWORK

In this paper, we bridge the two spaces via class prototypes in both the semantic and visual spaces, i.e., a semantic prototype $\boldsymbol{P}_y^s$ and a visual prototype $\boldsymbol{P}_y^v$ for each class $y$. In order to generate high-quality prototypes, we develop a learnable scheme called "*Isometric Propagation Network*

(IPN)" to iteratively refine the prototypes by propagating them on a graph of classes so each prototype is dynamically determined by the samples and attributes of its related classes.

In this section, we will first introduce the architecture of IPN. An illustration of IPN is given in Figure 2. Then, we will elaborate the episodic training strategy, in which a consistency loss is minimized for isometric alignment between the two spaces.

## 4.1 PROTOTYPE INITIALIZATION

Given a task $T$ defined on a subset of classes $\mathcal{Y}^T$ with training data $\mathcal{D}^T$, IPN first initializes the semantic and visual prototype for each class $y \in \mathcal{Y}^T$. The semantic prototype of class $y$ is transformed from its attributes $\boldsymbol{s}_y$. Specifically, we have drawn on the expert modules from (Zhang & Shi, 2019) to initialize the semantic prototypes $\boldsymbol{P}_y^s[0]$. In previous works (Snell et al., 2017; Santoro et al., 2016), the visual prototype of a class is usually computed by averaging the features of images from the class. In IPN, for each seen class $y \in \mathcal{Y}^T \cap \mathcal{Y}^{seen}$, we initialize its visual prototype $\boldsymbol{P}_y^v[0]$ by

$$\boldsymbol{P}_y^v[0] = \frac{\boldsymbol{W}}{|\mathcal{D}_y^T|} \sum_{(\boldsymbol{x}_i, y_i) \in \mathcal{D}_y^T} f_{cnn}(\boldsymbol{x}_i), \ \ D_y^T \triangleq \{(\boldsymbol{x}_i, y_i) \in \mathcal{D}^T : y_i = y\}, \tag{1}$$

where $f_{cnn}(\cdot)$ is a backbone convolutional neural nets (CNN) extracting features from raw images, and $\boldsymbol{W}$ is a linear transformation aiming to change the dimension of the visual prototype to be the same as semantic prototypes. We argue that $\boldsymbol{W}$ is necessary because (1) the visual feature produced by the CNN is usually in high dimension, and reducing its dimension can save significant computations during propagation; (2) keeping the dimensions of the visual and semantic prototypes the same can avoid the dominance of either of them in the model.

During test, we initialize the visual prototype of each unseen class $y \in \mathcal{Y}^T \cap \mathcal{Y}^{unseen}$ as the weighted sum of its $K$-nearest classes from $\mathcal{Y}^{seen}$ in the semantic space[1], i.e.,

$$\boldsymbol{P}_y^v[0] = \boldsymbol{W} \cdot \sum_{z \in \mathcal{N}_y} a^s(\boldsymbol{P}_y^s[0], \boldsymbol{P}_z^s[0]) \cdot \boldsymbol{P}_z^v, \tag{2}$$

where $\boldsymbol{W}$ is the same transformation from Eq. (1), and $\mathcal{N}_y$ is the set of the top-$K$ ($K$ is a hyperparameter) seen classes[2] with the largest similarities $c^s(\boldsymbol{P}_y^s[0], \boldsymbol{P}_z^s[0])$ to class $y$ in semantic space.

## 4.2 CATEGORY GRAPH GENERATION

Although the propagation needs to be conducted on a category graph, IPN does not rely on any given graph structure as many previous works (Kampffmeyer et al., 2019; Wang et al., 2018) so it has less limitations when applied to more general scenarios. In IPN, we inference the category graph of $\mathcal{Y}^T$ by applying thresholding to the attention scores between the initialized prototypes. In the visual space, the set of edges is defined by

$$\mathcal{E}^{T,v} = \left\{ (y, z) : y, z \in \mathcal{Y}^T, c^v(\boldsymbol{P}_y^v[0], \boldsymbol{P}_z^v[0]) \geq \epsilon \right\}, \tag{3}$$

where $\epsilon$ is the threshold. Thereby, the category graph is generated as $\mathcal{G}^{T,v} = (\mathcal{Y}^T, \mathcal{E}^{T,v})$. Similarly, in the semantic space, we generate another graph $\mathcal{G}^{T,s} = (\mathcal{Y}^T, \mathcal{E}^{T,s})$.

## 4.3 ISOMETRIC PROPAGATION

Given the initialized prototypes and generated category graphs in the two spaces, we apply the following prototype propagation in each space for $\tau$ steps in order to refine the prototype of each class. In the visual space, each propagation step updates

$$\boldsymbol{P}_y^v[t+1] = \sum_{z:(y,z) \in \mathcal{E}^{T,v}} a^v(\boldsymbol{P}_y^v[t], \boldsymbol{P}_z^v[t]) \cdot \boldsymbol{P}_z^v[t], \tag{4}$$

---

[1]This only happens in the test since we do not use any information of unseen classes during training.
[2]Here the notation $\mathcal{N}_y$ needs a little overloading since it denotes the set of neighbor classes of $y$ elsewhere in the paper.

In the semantic space, we apply similar propagation steps over classes on graph $\mathcal{G}^{T,s}$ by using the attention module associated with $a^s(\cdot, \cdot)$. Intuitively, the attention module computes the similarity between the two prototypes. In the visual space, given the prototypes $\boldsymbol{P}_y^v$ and $\boldsymbol{P}_z^v$ for class $y$ and $z$, the attention module first computes the following cosine similarity:

$$c^v(\boldsymbol{P}_y^v, \boldsymbol{P}_z^v) = \frac{\langle h^v(\boldsymbol{P}_y^v), h^v(\boldsymbol{P}_z^v) \rangle}{\|h^v(\boldsymbol{P}_y^v)\|_2 \cdot \|h^v(\boldsymbol{P}_z^v)\|_2}, \tag{5}$$

where $h^v(\cdot)$ is a learnable transformation and $\| \cdot \|_2$ represents the $\ell_2$ norm. For a class $y$ on a category graph, we can use the attention module to aggregate messages from its neighbor classes $\mathcal{N}_y$. In this case, we need to further normalize the similarities of $y$ to each $z \in \mathcal{N}_y$ as follows.

$$a^v(\boldsymbol{P}_y^v, \boldsymbol{P}_z^v) = \frac{\exp[\gamma c^v(\boldsymbol{P}_y^v, \boldsymbol{P}_z^v)]}{\sum_{z \in \mathcal{N}_y} \exp[\gamma c^v(\boldsymbol{P}_y^v, \boldsymbol{P}_z^v)]}, \tag{6}$$

where $\gamma$ is a temperature parameter controlling the smoothness of the normalization. In the semantic space, we have similar definitions of $h^s(\cdot)$, $c^s(\cdot, \cdot)$ and $a^s(\cdot, \cdot)$. In the following, we will use $\boldsymbol{P}_y^v[t]$ and $\boldsymbol{P}_y^s[t]$ to represent the visual and semantic prototype of class $y$ at the $t^{th}$ step of propagation. We will use $|\cdot|$ to represent the size of a set and $\mathrm{ReLU}(\cdot) \triangleq \max\{\cdot, 0\}$.

After $\tau$ steps of propagation, we concatenate the prototypes achieved in the two spaces for each class to form the final prototypes:

$$\boldsymbol{P}_y = [\boldsymbol{P}_y^v[\tau], \boldsymbol{P}_y^s[\tau]], \ \ \forall \, y \in \mathcal{Y}^T. \tag{7}$$

The formulation of $a^s(\cdot, \cdot)$ and $c^s(\cdot, \cdot)$ follows the same way as $a^v(\cdot, \cdot)$ and $c^v(\cdot, \cdot)$. To be simple, we don't rewrite the formulations here.

### 4.4 Classifier generated from Prototypes

Given the final prototypes and a query/test image $\boldsymbol{x}$, we apply a fully-connected layer to predict the similarity between $\boldsymbol{x}$ and each class's prototype. For each class $y \in \mathcal{Y}^T$, it computes

$$f(\boldsymbol{x}, \boldsymbol{P}_y) = \boldsymbol{w}\,\mathrm{ReLU}(\boldsymbol{W}^{(1)}\boldsymbol{x} + \boldsymbol{W}^{(2)}\boldsymbol{P}_y + \boldsymbol{b}^{(1)}) + b, \tag{8}$$

Intuitively, the above classifier can be interpreted as a distance classifier with distance metric $f(\cdot, \cdot)$. When applied to a query image $\boldsymbol{x}$, the predicted class is the most probable class, i.e., $\hat{y} = \arg\max_{y \in \mathcal{Y}^T} \Pr(y|\boldsymbol{x}; \boldsymbol{P})$, where $\Pr$ denotes the probability.

### 4.5 Training Strategies

**Episodic training on subgraphs.** In each episode, we sample a subset of training classes $\mathcal{Y}^T \subseteq \mathcal{Y}^{seen}$, and then build a training (or support) set $\mathcal{D}_{train}^T$ and a validation (or query) set $\mathcal{D}_{valid}^T$ from the training samples of $\mathcal{Y}^T$. They are two disjoint sets of samples from the same subset of training classes. We *only use* $\mathcal{D}_{train}^T$ to generate the prototypes $\boldsymbol{P}$ by the aforementioned procedures, and then apply backpropagation to minimize the loss of $\mathcal{D}_{valid}^T$. This equals to an empirical risk minimization, i.e.,

$$\min \sum_{\mathcal{Y}^T \sim \mathcal{Y}^{seen}} \sum_{(\boldsymbol{x}, y) \sim \mathcal{D}_{valid}^T} -\log \Pr(y|\boldsymbol{x}; \boldsymbol{P}), \tag{9}$$

This follows the same idea of meta-learning Ravi & Larochelle (2016): it aims to minimize the validation loss instead of the training loss and thus can further improve generalization and restrain overfitting. Moreover, since we train the IPN model on multiple random tasks $T$ and optimize the propagation scheme over different subgraphs (each generated for a random subset of classes), the generalization capability of IPN to unseen classes and new tasks can be substantially enhanced by this meta-training strategy. Moreover, it effectively reduces the propagation costs since computing the attention needs $O(|\mathcal{Y}^T|^2)$ operations. In addition, as shown in the experiments, episodic training is effective in mitigating the problem of data imbalance and overfitting to seen classes, which harasses many ZSL methods.

**Consistency loss for alignment of visual and semantic space.** In ZSL, the interaction and alignment between different spaces can provide substantial information about class dependency and is critical to the generation of accurate class prototypes. In IPN, we apply prototype propagation in two separate spaces on their own category graphs by using two untied attention modules. In order to keep the propagation in the two spaces consistent, we add an extra consistency loss to the training objective in Eq. (9). It computes the KL-divergence between two distributions defined in the two spaces for each class $y \in \mathcal{Y}^T$ at every propagation step $t$:

$$D_{KL}(p_y^v[t]||p_y^s[t]) = - \sum_{z \in \mathcal{Y}^T} p_y^v[t]_z \cdot \log \frac{p_y^s[t]_z}{p_y^v[t]_z}, \qquad (10)$$

where

$$p_y^v[t]_z \triangleq \frac{\exp[c^v(\boldsymbol{P}_y^v[t], \boldsymbol{P}_z^v[t])]}{\sum_{z \in \mathcal{Y}^T} \exp[c^v(\boldsymbol{P}_y^v[t], \boldsymbol{P}_z^v[t])]}, \qquad (11)$$

$$p_y^s[t]_z \triangleq \frac{\exp[c^s(\boldsymbol{P}_y^s[t], \boldsymbol{P}_z^s[t])]}{\sum_{z \in \mathcal{Y}^T} \exp[c^s(\boldsymbol{P}_y^s[t], \boldsymbol{P}_z^s[t])]}. \qquad (12)$$

The consistency loss enforces the two attention modules in the two spaces to generate the same attention scores over classes, even when the prototypes and attention module's parameter are different in the two spaces. In this way, they share similar class dependency over the propagation steps. So the generated dual prototypes are well aligned and can provide complementary information when generating the classifier. The final optimization we used to train IPN becomes:

$$\min \sum_{\mathcal{Y}^T \sim \mathcal{Y}^{seen}} \left[ \sum_{(\boldsymbol{x}, y) \sim \mathcal{D}_{valid}^T} -\log \Pr(y|\boldsymbol{x}; \boldsymbol{P}) + \lambda \sum_{t=1}^{\tau} \sum_{y \in \mathcal{Y}^T} D_{KL}(p_y^v[t]||p_y^s[t]) \right], \qquad (13)$$

where $\lambda$ is weight for the consistency loss, and $\boldsymbol{P}$, $p_y^v[t]$, and $p_y^s[t]$ are computed only based on $\mathcal{D}_{train}^T$, the training (or support) set of task-$T$ for classes in $\mathcal{Y}^T$.

## 5 EXPERIMENTS

We evaluate IPN and compare it with several state-of-the-art ZSL models in both ZSL setting (where the test set is only composed of unseen classes) and generalized ZSL setting (where the test set is composed of both seen and unseen classes). We report three standard ZSL evaluation metrics, and provide ablation study of several variants of IPN in the generalized ZSL setting. The visualization of the learned class representation is shown in Figure 1.

### 5.1 DATASETS AND EVALUATION CRITERIA

**Small and Medium benchmarks.** Our evaluation includes experiments on three standard benchmark datasets widely used in previous works, i.e., AWA2 (Xian et al., 2019a), CUB (Welinder et al., 2010) and aPY (Farhadi et al., 2009). The former two are extracted from ImageNet-1K, where AWA2 contains 50 animals classes and CUB contains 200 bird species. aPY includes 32 classes. Detailed statistics of them can be found in Table 1.

**Two new large-scale benchmarks.** The three benchmark datasets above might be too small for modern DNN models since the training may suffers from large variance and end-to-end training usually fails. Previous ZSL methods therefore fine-tune a pre-trained CNN to achieve satisfactory performance on these datasets, which is costly and less practical. Given these issues, we proposed two new large-scale datasets extracted from *tiered*ImageNet (Ren et al., 2018), i.e., *tiered*ImageNet-Segregated and *tiered*ImageNet-Mixed, whose statistics are given in Table 1. Comparing to the three benchmarks, they are over $10\times$ larger, and training a ResNet101 (He et al., 2016) from scratch on their training sets can achieve a validation accuracy of $65\%$, indicating that end-to-end training without fine-tuning is possible. These two benchmarks differ in that the "Mixed" version allows a seen class and an unseen class to belong to the same super category, while the "Segregated" version does not. The class attributes are the word embeddings of the class names provided by GloVe (Pennington et al., 2014). We removed the classes that do not have embeddings in GloVe.

Following the most popular setting of ZSL (Xian et al., 2019a), we report the per-class accuracy and harmonic mean of the accuracy on seen and unseen classes.

| Dataset | #Attributes | #Seen | #Unseen | #Imgs(Tr-S) | #Imgs(Te-S) | #Imgs(Te-U) |
|---|---|---|---|---|---|---|
| CUB (Welinder et al., 2010) | 312 | 150 | 50 | 7,057 | 1,764 | 2,967 |
| AWA2 (Xian et al., 2019a) | 85 | 40 | 10 | 23,527 | 5,882 | 7,913 |
| aPY (Farhadi et al., 2009) | 64 | 20 | 12 | 5,932 | 1,483 | 7,924 |
| *tiered*ImageNet-Segregated | 300 | 422 | 182 | 378,884 | 162,952 | 232,129 |
| *tiered*ImageNet-Mixed | 300 | 447 | 157 | 399,741 | 171,915 | 202,309 |

Table 1: Datasets Statistics. CUB, AWA2 and aPY are three standard ZSL benchmarks, while *tiered*ImageNet-Segregated/Mixed are two new large-scale ones proposed by us with different discrepancy between seen and unseen classes. "Tr-S", "Te-S" and "Te-U" denote seen classes in training, seen classes in test and unseen classes in test.

| Methods | | CUB | | | AWA2 | | | aPY | | | *tiered*-M | | | *tiered*-S | | |
|---|---|---|---|---|---|---|---|---|---|---|---|---|---|---|---|---|
| | | S | U | H | S | U | H | S | U | H | S | U | H | S | U | H |
| **Generative Models** | GDAN | **66.7** | 39.3 | 49.5 | 67.5 | 32.1 | 43.5 | **75.0** | 30.4 | 43.4 | 50.5 | 2.1 | 4.0 | 3.9 | 1.2 | 1.8 |
| | LisGAN | 57.9 | 46.5 | 51.6 | 76.3 | 52.6 | 62.3 | 68.2 | 34.3 | 45.7 | 1.6 | 0.1 | 0.2 | 0.2 | 0.1 | 0.1 |
| | CADA-VAE | 53.5 | 51.6 | 52.4 | 75.0 | 55.8 | 63.9 | - | - | - | **60.1** | 3.5 | 6.5 | 0.8 | 0.1 | 0.2 |
| | f-VAEGAN-D2 | 60.1 | 48.4 | 53.6 | 70.6 | 57.6 | 63.5 | - | - | - | - | - | - | - | - | - |
| | FtFT | 54.8 | 47.0 | 50.6 | 72.6 | 55.3 | 62.6 | - | - | - | - | - | - | - | - | - |
| **Discriminative Models** | Relation Net | 61.1 | 38.1 | 47.0 | **93.4** | 30.0 | 45.3 | - | - | - | 31.4 | 3.4 | 6.1 | 45.8 | 1.2 | 2.3 |
| | GAFE | 52.1 | 22.5 | 31.4 | 78.3 | 26.8 | 40.0 | 68.1 | 15.8 | 25.7 | - | - | - | - | - | - |
| | PQZSL | 51.4 | 43.2 | 46.9 | - | - | - | 64.1 | 27.9 | 38.8 | - | - | - | - | - | - |
| | MLSE | 71.6 | 22.3 | 34.0 | 83.2 | 23.8 | 37.0 | 74.3 | 12.7 | 21.7 | - | - | - | - | - | - |
| | CRNet | 56.8 | 45.5 | 50.5 | 78.8 | 52.6 | 63.1 | 68.4 | 32.4 | 44.0 | 58.0 | 2.5 | 4.9 | **57.6** | 1.0 | 2.0 |
| | TCN | 52.0 | 52.6 | 52.3 | 65.8 | 61.2 | 63.4 | 64.0 | 24.1 | 35.1 | - | - | - | - | - | - |
| | RGEN | 73.5 | 60.0 | 66.1 | 76.5 | 67.1 | 71.5 | 48.1 | 30.4 | 37.2 | - | - | - | - | - | - |
| | APN | 55.9 | 48.1 | 51.7 | 83.9 | 54.8 | 66.4 | 74.7 | 32.7 | 45.5 | - | - | - | - | - | - |
| | IPN (ours) | **73.8** | **60.2** | **66.3** | 79.2 | **67.5** | **72.9** | 66.0 | **37.2** | **47.6** | 50.1 | **6.1** | **11.0** | 54.4 | **2.5** | **4.7** |

Table 2: Performance of existing ZSL models and IPN on five datasets (generalized ZSL setting), where "S" denotes the per-class accuracy (%) on seen classes, "U" denotes the per-class accuracy (%) on unseen classes and "H" denotes their harmonic mean.

## 5.2 IMPLEMENTATION AND HYPERPARAMETER DETAILS

For a fair comparison, we follow the setting in (Xian et al., 2019a) and use a pre-trained ResNet-101 (He et al., 2016) to extract 2048-dimensional image features without fine-tuning. For the three standard benchmarks, the model is pre-trained on ImageNet-1K. For the two new datasets, it is pre-trained on the training set used to train ZSL models.

All hyperparameters were chosen on the validation sets provided by Xian et al. (2019a). We use the same hyperparameters tuned on AWA2 for other datasets since they are of similar data type, except for aPY on which we choose learning rate $1.0 \times 10^{-3}$ and weight decay $5.0 \times 10^{-4}$. For the rest of the datasets, we train IPN by Adam (Kingma & Ba, 2015) for 360 epochs with a weight decay factor $1.0 \times 10^{-4}$. The learning rate starts from $2.0 \times 10^{-5}$ and decays by a multiplicative factor 0.1 every 240 epochs. In each epoch, we update IPN on multiple $N$-way-$K$-shot tasks ($N$-classes with $K$ samples per class), each corresponding to an episode mentioned in Section 4.5. In our experiments, the number of episodes in each epoch is $n/NK$, where $n$ is the total size of the training set, $N = 30$ and $K = 1$ (a small $K$ can mitigate the imbalance between seen and unseen classes); $h^v$ and $h^s$ are linear transformations; we set temperature $\gamma = 10$, threshold $\epsilon = \cos 40^o$ for the cosine similarities between class prototypes, and weight for consistency loss $\lambda = 1$. We use propagation steps $\tau = 2$.

## 5.3 MAIN RESULTS

The results are reported in Table 2. Although most previous works (Xian et al., 2019b;a; Sariyildiz & Cinbis, 2019) show that the more complicated generative models usually outperforms the discriminative ones. Our experiments imply that IPN, as a simple discriminative model, can significantly outperform the generative models requiring much more training efforts. Most baselines achieve a

high $ACC_{seen}$ but much lower $ACC_{unseen}$ since they suffer from and more sensitive to the imbalance between seen and unseen classes. In contrast, IPN always achieves the highest H-metric among all methods, indicating less overfitting to seen classes. This is due to the more accurate prototypes achieved by iterative propagation in two interactive spaces across seen and unseen classes, and the episodic training strategy, which effectively improves the generalization.

In scaling the experiments to the two large datasets, discriminative models perform better than generative models, which are easily prone to overfitting or divergence. Moreover, during the test stage, generative models require to synthesize data for unseen classes, which may lead to expensive computations when applied to tasks with many classes. In contrast, IPN scales well to these large datasets, outperforming state-of-the-art baselines by a large margin. The performance of most models on *tiered*ImageNet-Segregated is generally worse than *tiered*ImageNet-Mixed because that the unseen classes in the former do not share the same ancestor classes with the seen classes, while in the latter such sharing is allowed.

## 5.4 ABLATION STUDY

**Are two prototypes necessary?** We compare IPN with its two variants, each using only one out of the two types of prototypes from the two spaces when generating the classifier. The results are shown in Figure 3. It shows that semantic prototype (only) performs much better than visual prototype (only), especially on the unseen classes. This is because that the unseen classes' semantic embedding is given while its visual information is missing (zero images) during the test stage. Although IPN is capable of generating a visual prototype based on other classes' images and class dependency, it is much less accurate than the semantic prototype. However, it also shows that the visual prototypes (even the ones of unseen classes) can provide extra information and further improve the performance of the semantic prototype, e.g., they improve the unseen classes' accuracy from 61.1% to 67.5%. One possible reason is that the visual prototype bridges the query image's visual feature with the semantic features of classes, and thus provides an extra "visual context" for the localization of the correct class. In addition, we observe that IPN with only the visual prototype converges faster than IPN with only a semantic prototype (30 epochs vs. 200 epochs). This might be an artifact caused by the pre-trained CNN, which already produces stable visual prototypes since the early stages. However, it raises a caveat that heavily relying on a pre-trained model might discourage the learning and results in a bad local optimum.

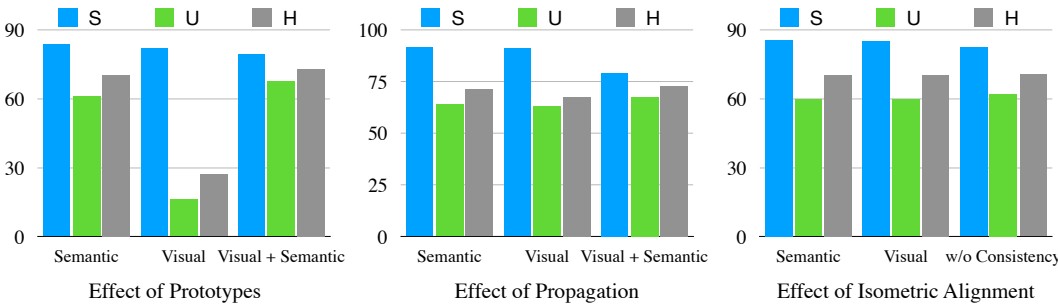

Figure 3: Ablation study of IPN on AWA2 (generalized ZSL setting). "S", "U" and "H" are the same notations used in Table 2. (a) Only one of the dual prototypes are used to generate classifiers. (b) Only one or none of the propagation in the two spaces are applied in IPN. (c) Different types of interaction between spaces. "visual/semantic attention only" means we use visual/semantic attention in both spaces, "two separate attentions" refers to IPN trained without consistency loss.

**How's the benefit of propagation?** We then evaluate the effects of the propagation scheme in the two spaces. In particular, we compare IPN with its variants that apply the propagation steps in only one of the two spaces, and the variant that does not apply any propagation after initialization. Their performance is reported in Figure 3. Without any propagation, the classification performance heavily relies on pre-trained CNN. Hence, the accuracy of seen classes can be much higher than that of the unseen classes. Compared to IPN without any propagation, applying propagation in either the visual space or the semantic space can improve the accuracy of unseen classes by 2∼3%, while the accuracy on the seen classes stays almost the same. When equipped with dual propagation, we

observe a significant improvement of the accuracy of the unseen classes but a degradation on the seen classes. However, they lead to the highest H-metric, which indicates that the dual propagation achieves a desirable trade-off between overfitting on seen classes and generalization to unseen classes.

**Effect of Isometric Alignment** In IPN, we minimize a consistency loss during training for alignment of the visual and semantic space. Here, we consider two other options for alignment that exchange the attention modules used in the two spaces, i.e., using the visual attention module in the semantic space, or using the semantic attention module in the visual space. We also include the IPN trained without consistency loss or any alignment as a baseline. Their results are reported in Figure 3. Compared to IPN with/without consistency loss, sharing a single attention module between the two spaces suffers from the overfitting problem on the seen classes. This is because that the visual features and semantic embeddings are two different types of features so they should not share the same similarity metric. In contrast, the consistency loss only enforces the output similarity scores of the two attention modules to be close, while the models/functions computing the similarity can be different. Therefore, to produce the same similarity scores, it does not require each class having the same features in the two spaces. Hence, it leads to a much softer regularization.

# 6 SENSITIVITY OF HYPERPARAMETERS

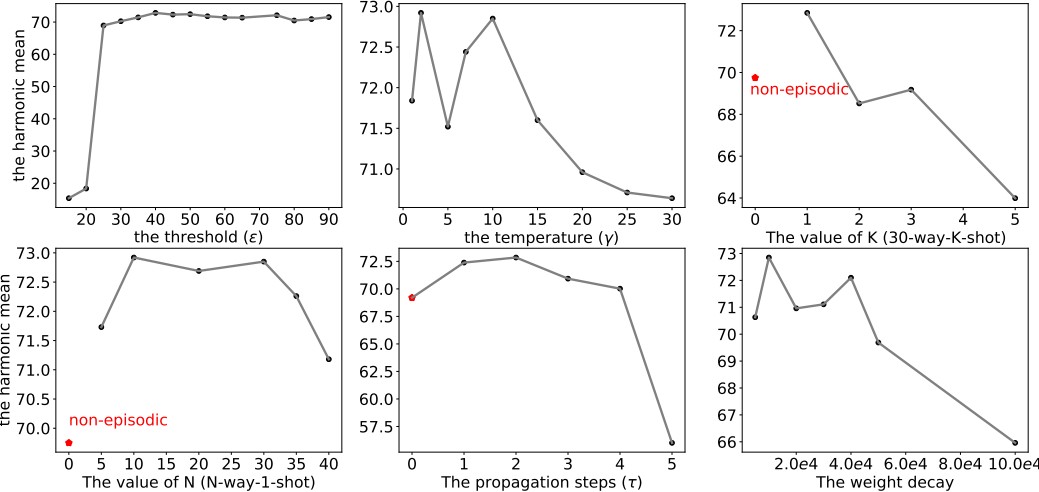

Figure 4: Performance analysis when using different values of hyperparameters on AWA2.

The proposed IPN has some hyperparameters as mentioned in Section 5.2. To analyze the sensitivity of those hyperparameters, such as threshold, temperature, N and K for N-way-K-shot task, propagation steps, etc. We try different values and show the performance on the AWA2 dataset in Figure 4. The temperature $\gamma$ is not sensitive for the accuracy. The threshold $\epsilon$ needs to be bigger than cosine($30^o$) to give IPN some space for the filtering. The propagation steps $\tau$ is best at 2. Weight decay is a sensitive hyperparameter and this is also found in previous ZSL literature (Zhang & Shi, 2019; Liu et al., 2019d).

# 7 CONCLUSIONS

In this work, we propose an Isometric Propagation Network, which learns to relate classes within each space and across two spaces. IPN achieves state-of-the-art performance and we show the importance of each component and visualize the interpretable relationship of the learned class representations. In future works, we will study other possibilities to initialize the visual prototypes of unseen classes in IPN.

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

## A  EVALUATION CRITERIA

Considering the imbalance between test classes, by following the most recent works (Xian et al., 2019a), given a test set $\mathcal{D}$, we evaluate ZSL methods' performance based on the per-class accuracy averaged over a targeted set of classes $\mathcal{Y}$, i.e.,

$$ACC_{\mathcal{Y}} = \frac{1}{|\mathcal{Y}|} \sum_{y \in \mathcal{Y}} \frac{1}{|D_y|} \sum_{(\boldsymbol{x}, y) \in D_y} \mathbb{1}[\widehat{y} = y], \tag{14}$$

$$D_y \triangleq \{(\boldsymbol{x}_i, y_i) \in \mathcal{D} : y_i = y\}.$$

Following Xian et al. (2019a), we report $ACC_{\mathcal{Y}}$ when $\mathcal{Y} = \mathcal{Y}_{seen}$ and $\mathcal{Y} = \mathcal{Y}_{unseen}$, and the harmonic mean of them, i.e.,

$$H = \frac{2 \cdot ACC_{\mathcal{Y}_{seen}} \cdot ACC_{\mathcal{Y}_{unseen}}}{ACC_{\mathcal{Y}_{seen}} + ACC_{\mathcal{Y}_{unseen}}}. \tag{15}$$

| | Methods | CUB | AWA2 | aPY | *tiered*-M | *tiered*-S |
|---|---|---|---|---|---|---|
| **Generative Models** | LisGAN (Li et al., 2019a) | 58.8 | 70.6 | 43.1 | 1.6 | 0.1 |
| | f-VAEGAN-D2 (Xian et al., 2019b) | 61.0 | 71.1 | - | - | - |
| **Discriminative Models** | SAE (Kodirov et al., 2017) | 33.3 | 54.1 | 8.3 | - | - |
| | DEM (Zhang et al., 2017) | 51.7 | 67.1 | 35.0 | - | - |
| | RN (Sung et al., 2018) | 55.6 | 64.2 | 30.1 | 6.1 | 2.5 |
| | GAFE (Liu et al., 2019d) | 52.6 | 67.4 | **44.3** | - | - |
| | **IPN (ours)** | **59.6** | **74.4** | 42.3 | **14.0** | **5.0** |

Table 3: Per-class accuracy (%) on unseen classes achieved by existing ZSL models and IPN on five datasets (the ZSL setting).

## B   EXPERIMENTS ON ZERO-SHOT LEARNING SETTING

The comparison between IPN to other baselines on the setting of Zero-shot Learning is shown in Table 3. On CUB, AWA2, aPU, *tiered*-**M**, and *tiered*-**S**, our IPN outperforms other algorithms by a large margin.

## C   MORE EXPERIMENTS ON LARGE-SCALE DATASETS

In our work, we follow the train/test split suggested by Frome et al. (2013), who proposed to use the 21K ImageNet dataset for zero-shot evaluation. Here, 2-hops and 3-hops represent that the evaluation is over the classes which are 2/3 hops away from ImageNet-1k. Different from experiments on AWA2, we use $N$=20 and $K$=5 for the full ImageNet. Besides, since the edges between each class are provided on the full ImageNet, we directly use them instead of calculating by Eq. (3).

| Methods | Hit@k (%) **[2-hops]** | | | | Hit@k (%) **[3-hops]** | | | |
|---|---|---|---|---|---|---|---|---|
| | 1 | 5 | 10 | 20 | 1 | 5 | 10 | 20 |
| ConSE (Norouzi et al., 2013) | 8.3 | 21.8 | 30.9 | 41.7 | 2.6 | 7.3 | 11.1 | 16.4 |
| SYNC (Changpinyo et al., 2016) | 10.5 | 28.6 | 40.1 | 52.0 | 2.9 | 9.2 | 14.2 | 20.9 |
| EXEM (Changpinyo et al., 2017) | 12.5 | 32.3 | 43.7 | 55.2 | 3.6 | 10.7 | 16.1 | 23.1 |
| GCNZ (Wang et al., 2018) | 19.8 | 53.2 | 65.4 | 74.6 | 4.1 | 14.2 | 20.2 | 27.7 |
| DGP (Kampffmeyer et al., 2019) | 26.6 | 60.3 | 72.3 | 81.3 | 6.3 | 19.3 | 27.7 | 37.7 |
| **IPN (ours)** | **27.1** | **61.1** | **73.8** | **82.9** | **6.8** | **20.1** | **28.9** | **39.6** |

Table 4: Top-k accuracy for the different models on the ImageNet dataset. Accuracy when only testing on unseen classes. Results are taken from (Kampffmeyer et al., 2019).

