# OpenReview forum: "Isometric Propagation Network for Generalized Zero-shot Learning"
_ICLR.cc/2021/Conference — ICLR 2021 Poster_

### Official Review · AnonReviewer3 · 2020-10-26
**Please find below for the detailed comments**

**Rating:** 4
**Confidence:** 4

**Review:**

This paper proposes an Isometric Propagation Network (IPN) for ZSL. The proposed IPN aims to learn to strengthen the relation between classes within each space and align the class dependency in the two spaces. IPN achieves the state-of-the-art performance on three popular ZSL benchmarks.

Strengths:
1. The explanation of the objective function and the main formula is clear.
2. The introduction of related work is very organized.

Weaknesses:
1. The motivation and the advantages of IPN are unclear.
2. Most compared methods are out of data. Some generative or discriminative ZSL methods proposed in CVPR’20 or ECCV’20 are not compared.
3. More experimental results besides ZSL results and H value should be given to show the advantages of IPN.

---

> ### Author Response · Authors · 2020-11-11
> **Would you mind to elaborate with more details of them?**
>
> Thanks for your comments!
>
> We notice that you have briefly listed three weaknesses but we need more details to provide you a better response. Would you mind to elaborate with more details of them?
>
> **1**. The motivation and advantages of IPN are unclear.
>
>   Can you specify which aspects of motivation and which advantages you found unclear? Our main statements of motivation and advantages can be found in Section 1, and we can give a shorter summary here: To resolve the problem of imbalanced supervision from the semantic and the visual space, we propose Isometric Propagation Network (IPN), which learns to strengthen the relation between classes within each space and align the class dependency in the two spaces. In contrast to only aligning the resulted static representation, we regularize the two *dynamic* propagation *procedures* to be isometric.
>
> **2**. Most compared methods are out of data. Some generative or discriminative ZSL methods proposed in CVPR’20 or ECCV’20 are not compared.
>
>   We have a baseline RGEN [a] from CVPR2020 in Table 1. Could you specify which papers from CVPR’20 or ECCV’20 we need to compare with?
>
> **3**. More experimental results besides ZSL results and H value should be given to show the advantages of IPN.
>
>   This paper targets on ZSL problem and H metric is the most popular evaluation metric in this field [b]. Which specific ZSL evaluation metrics do you refer to?
>
> [a] Region graph embedding network for zero-shot learning
>
> [b] Zero-shot learning a comprehensive evaluation of the good, the bad and the ugly

---

### Official Review · AnonReviewer1 · 2020-10-28
**Interesting interaction between visual and semantic prototypes with good results; writing needs to be improved**

**Rating:** 6
**Confidence:** 3

**Review:**

**Summary of paper**
In this paper Zero Shot Classification is studied using prototypes. Each class is represented with a visual and semantic prototype, and at test time compared to a visual example + prototype for a(n unseen) test class. The most similar test class is chosen. In this work a novel method is proposed to construct the prototypes, which are trained in an episode learning setting. On various benchmarks the proposed method performs better than existing method for the generalized zero-shot classification task (seen + unseen) test classes.


**Strengths**
1. This paper combines semantic and visual prototypes via a kind of attention method for (unseen) test classes (Eq 2), and at train time via a KL divergence, yielding more visual&semantic prototype neighbours. That is interesting way of combining the two spaces. Could the attention module directly use both spaces, and not only via a KL loss?

2. The proposed method obtains good results on different benchmark tasks.

**Weaknesses**
1. Notation is not always clear, and partly undefined. Below a list of some examples:
  * Eq 1: given that D is fixed for a class (at any iteration, not only at [0]), this seems to be the simple projection: W P, where W depends on the iteration, so may be W[0] should be used. Note that is unclear why the visual prototypes should be mapped to the semantic prototypes. The reverse is equally explainable. [Please make clear that W is shared among all prototypes].
  * Eq 2 (and below): a^s and c^s are not defined.
  * Eq 2: How is P^v_z defined? When similar to Eq 1, then there is a double projection with W.
  * Eq 8: If w and b are not class dependent (ie depend on Py), then at least b do not add anything to the argmax.
  * Pr (below Eq 8) is undefined.
  * Section 4.5: unclear what parameters are trained here (W, W1, W2, b1, w, b)? Unclear how important the episode training is compared to 'standard' training.
2. Relation to other methods are unclear. For example, the label/graph propagation is not related to other methods. The episodic learning is not related to other method (in ZSL), also its importance is only implicitly visible from the experimental results (by comparing the ablations with results from the main table).
3. Unjustified claims and missing experiments:
 * A new large scale benchmark to train end-to-end is introduced [Section 5.1 paragraph 'Two new large-scale benchmarks'], but in 5.2 it is stated 'we use pre-trained classifiers without fine-tuning'. Besides many more papers on ZSL have conducted experiments on (subsets of) ImageNet.
  * Claim in Sect 5.3 'For example IPN brings 10% improvement to H-metric on AWA2', according to Table 1: 72.9 vs 71.5. That is a 1% improvement.
  *  'Initialisation of visual prototypes of unseen classes is important' [P7, Sect 5.3] (currently done using Eq 2). But it is left unclear how this is studied, concluded.
 * The method relies on a few hyper-parameters (temperature, threshold, episode learning N, K, learning weight (lambda), and the number of propagation steps), these are not studied, but for eg the propagation steps it is claimed that more steps perform better. This should be studied.
 * There are a few alternatives to the same idea (ie how to use the neighbour propagation in the two spaces) which are not explored.


 **Minor**
- The anonymous GitHub repository is not available (404 error)
- Figure 1: It seems that a nearest prototype classifier is used instead of the depicted nearest neighbour classifier
- Typo: Page 3, Sect 4, first sentence: 'we bridge the three spaces' --> two?
- Page 4, Sect 4.1: It is unclear what the geometry structure of  semantic prototypes are and how these are preserved.
- Page 8, Sect 6, first sentence: 'each space ans across two' --> and


**Conclusion**
This paper proposes a novel method to learn semantic and visual prototypes for zero-shot classification, which uses episodic learning. The prototypes are propagated over a visual/semantic neighbours graph. That is an interesting approach and experimentally the method obtains good results on various benchmarks. The influence of some of the key  parameters of the method are not studied, nor the disentanglement between the episodic training and the prototype propagation. This combined with the many (minor!) flaws and unclarities in the paper yields my rating of marginally below the acceptance threshold.

** Post Rebuttal**
The paper has been improved based on the reviews, I think this paper is now ok to accept.

---

> ### Author Response · Authors · 2020-11-15
> **Careful Proofread; More Results for Different Hyperparameters and Episodic Training**
>
> Thank you for the comprehensive reviews and detailed comments and suggestions!
> Here are our responses to your concerns:
>
> **Q1**: Notation is not always clear, and partly undefined.
>
> **A1**: Thanks for pointing these out! We have updated the paper to add more explanations.
>
> **Q2**: Relation to other methods are unclear. For example, the label/graph propagation is not related to other methods. The episodic learning is not related to other methods (in ZSL), also its importance is only implicitly visible from the experimental results (by comparing the ablations with results from the main table).
>
> **A2**: Please refer to the third paragraph ("Graph in ZSL") of Section 2 for the discussion of related works to graph/label propagation, and the last paragraph of Section 2 for episodic training.
> Episodic training experiment result: We have added the comparison of training in an episodic way or traditional supervised learning way in Figure 3 on Page 9. It shows more than 2.5% improvements on the H metric for AWA2.
>
> **Q3**: Claim in Sect 5.3 'For example IPN brings 10% improvement to H-metric on AWA2', according to Table 1: 72.9 vs 71.5. That is a 1% improvement.
>
> **A3**: Thanks for pointing out this issue. We have deleted this claim.
>
> **Q4**: 'Initialisation of visual prototypes of unseen classes is important' [P7, Sect 5.3] (currently done using Eq 2). But it is left unclear how this is studied, concluded.
>
> **A4**: For the current version, we just use the most natural way to initialize the visual prototypes of unseen classes. Intuitively, this initialization is important since visual prototypes serve as half of the inputs to the model but we don't have access to the images. In our revised version, we removed this claim from the experiment section but mentioned in Section 6 as future studies.
>
> **Q5**: The method relies on a few hyper-parameters (temperature, threshold, episode learning N, K, learning weight (lambda), and the number of propagation steps), these are not studied, but for eg the propagation steps it is claimed that more steps perform better. This should be studied.
>
> **A5**: Thanks for this valuable suggestion. We have added Section 6 and Figure 3 to show the sensitivity of these hyperparameters to the performance on AWA2, including the result when using a different number of propagation steps. For example, the temperature is not sensitive to accuracy. The threshold needs to be bigger than cosine(30). Please find more details in Section 6 in our revised version.
>
> **Q6**: The anonymous GitHub repository is not available (404 error)
>
> **A6**: We have fixed this issue, please see code at: https://anonymous.4open.science/r/041f3669-663e-4048-9811-9faa620cc551/
>
> **Q7**: Figure 1: It seems that a nearest prototype classifier is used instead of the depicted nearest neighbour classifier
>
> **A7**: Thanks for this comment. Figure 1 just shows the embedding of the prototypes. The images in Figure 1 are just some illustration samples for a better understanding.
>
> **Q8**: Other Minor Issues.
>
> **A8**: We have fixed them in the updated version. Thanks for pointing them out!
>
> **Thank you for so many valuable suggestions. I hope you feel the flaws are addressed in our response and revised submission. Please let us know if you have any concerns.**

---

> > ### Comment · AnonReviewer1 · 2020-11-17
> > **About k-NN**
> >
> > Thanks for the update manuscript. I'll revisit that one soon.
> >
> > I want to re-raise the issue of the kNN classifier - as also raised by AR4. From the paper:
> > >Eq 8.
> > >Intuitively, the above classifier can be interpreted as a soft KNN classifier with similarity metric f(·, ·).
> >
> > For sure this classifier can be seen as a **distance** based classifier. However, AFAIK it is not a **kNN** classifier (then each of the training examples will yield a prototype, but rather a ***nearest class mean / nearest prototype*** kind of classifier.
> >
> > See for example Prototypical Networks [https://arxiv.org/abs/1703.05175]

---

> > > ### Author Response · Authors · 2020-11-18
> > > **Thank you!**
> > >
> > > Thank you for your comment on our response and your time to read our response in advance!
> > >
> > > We have revised it as a distance-based classifier in our submission to be more accurate.
> > >
> > > Please let us know if there's anything else you feel unclear about!
> > >
> > > Regards,
> > >
> > > Authors of submission #363

---

### Official Review · AnonReviewer4 · 2020-10-29
**Interesting idea but ablation studies are weak and raises many questions**

**Rating:** 7
**Confidence:** 4

**Review:**

This paper focuses on improving zero-shot classification by reducing the bias of the classifier towards seen classes. The bias occurs since the embedding is trained with visual examples from the seen classes, while using only the attribute information from unseen classes for testing. Authors propose an isometric propagation network that build a graph in both visual and semantic space, performs some steps of propagation, and then uses the updated prototypes for training a classifier. They use attention to construct the graph and also use attention to regularize the graph edges between the two spaces to be isometric. Authors also propose to use an episodic training method to improve learning.

Pros:
- Paper is well motivated and targets an important problem in ZSL
- Achieves SOTA results on standard benchmarks and proposed large scale benchmarks
- Show ablation studies for the proposed components
- Idea of using prototypes that are more informative than either semantic or visual attributes/features is interesting

Cons/Questions
- The paper could have been better written. It was hard to follow at some points
- Section 4: "In this paper, we bridge the three spaces via class... " Is it three or two spaces
- I was not convinced with the results in the ablation studies.  For most cases the harmonic mean is close the best value- which raises questions about how much these additional contributions are helping and what exactly is resulting in improvement over the baseline. The mode without any propagation achieves H=70.7 in table2(b). How do the method achieves 70.7 (which is more than most SOTA methods) without any propagation. Similarly the consistency loss only seems to be improving performance by 2.9% over H=70.1. Either these ablation studies are not set properly or the baseline is already high.
- How much is the episodic training helping?
- Can the authors provide a baseline the only uses the semantic features and then learning the classifier in Eq (8). Also, did the authors try using the prototypes in an embedding based classifier
- Why is the classifier in Eq (8) referred to as a KNN classifier.

Overall I find the idea interesting but I am giving a borderline rating due to several unanswered questions and lack of strong ablation studies highlighting the effectiveness of the proposed idea.

Final rating- I am satisfied with the author's response and updating my ratings.

---

> ### Author Response · Authors · 2020-11-15
> **More Empirical Results for Ablation Study**
>
> Thank you for reviewing our submission and the comprehensive comments! We are very happy to help to address your concerns!
>
> **Q1**: I was not convinced with the results in the ablation studies. For most cases, the harmonic mean is close to the best value, which raises questions about how much these additional contributions are helping and what exactly is resulting in improvement over the baseline. The mode without any propagation achieves H=70.7 in table2(b). How does the method achieve 70.7 (which is more than most SOTA methods) without any propagation? Similarly, the consistency loss only seems to be improving performance by 2.9% over H=70.1. Either these ablation studies are not set properly or the baseline is already high. Can the authors provide a baseline that only uses the semantic features and then learns the classifier in Eq (8)? Also, did the authors try using the prototypes in an embedding based classifier?
>
> **A1**: Thanks for this valuable suggestion! The previous ablations only show the importance of every component and still keep a part of our model, i.e., "without any propagation" still has two prototypes which is also one of the proposed modules in our submission. As suggested by you, the performance comparison with respect to two baselines are:
>
> | |only uses semantic features&nbsp;&nbsp;|prototypes with embedding based classifier&nbsp;&nbsp;|IPN|
> |:---:|:---:|:--:|:---:|
> | H metric&nbsp;&nbsp;  | 67.3   |  43.1 | **72.9** |
>
>
> **Q2**: How much is the episodic training helping?
>
> **A2**: Thanks for this comment! We have added this result in Figure 3 to clarify the gain of episodic training. It shows more than 2.5% improvement on H for the AWA2 dataset.
>
> **Q3**: Why is the classifier in Eq (8) referred to as a KNN classifier?
>
> **A3**: Thanks for your suggestion! We have revised it as a distance-based classifier with a distance metric $f(·,·)$.
>
>
> **We are pleased that you find our idea interesting, and hope our response answered your questions.**

---

### Official Review · AnonReviewer2 · 2020-10-29
**AnonReviewer2 [Edited after the author's response]**

**Rating:** 7
**Confidence:** 5

**Review:**

*Paper Summary*
The authors propose a novel computational pipeline to tackle a well-known problem in zero-shot learning: although multiple visual instances are available for the classes and categories to be recognized, one and only one semantic embedding is available to describe the classes/categories while using side information like attributes or relevant textual information. To cope with that problem, authors learn visual and semantic prototypes which are then adopted to perform gradient descent over a graph in which the topological relationship among similar/dissimilar classes are preserved. In the experimental validation, the proposed method shows its superiority among a number of prior methods in zero-shot learning, including discriminative and generative methods.

*Pros*
* The computational pipeline is novel and very original with respect to prior work, which mainly focuses on feature generating mechanisms
* Several state-of-the-art approaches are outperformed on 3 classical benchmark datasets for zero-shot learning (AWA2, CUB, aPY)

*Cons*
* **Sensitivity**. The method is depending upon a number of hyper-parameters (such as the threshold $\varepsilon$ for the visual/semantic neighbor graphs or the softmax temperature $\gamma$).
* **About datasets**. Although tiered-ImageNet is a popular benchmark in few-shot learning, there are methods which provide experimental results on the full ImageNet, such as [Kampffmetyer et al., Rethinking Knowledge Graph Propagation for Zero-Shot Learning, CVPR 2019]. It is not fully clear to which extent the proposed variants of tiered-ImageNet, adapted for zero-shot learning, would improve upon the existing ImageNet specs.

*Minor Comments*
The references (Xian et al. 2019a) and (Xian et al. 2019b) refer to the same paper.

*Pre-Rebuttal Evaluation* I think the method is novel and original, and it is adopting a very original direction which is dissimilar for mainstream approaches in zero-shot learning. My only questions relate to the computational sensitivity of the proposed method, which needs some further experimental shreds of evidence. Also, the datasets selected by authors need some clarifications, in both why a new benchmark was introduced and why. It would be great to have more experimental pieces of evidence on other benchmark datasets which are usually considered in zero-shot learning (such as the ''standard'' ImageNet or classical datasets such as SUN -- see (Xian et al. 2019a) -- or FLO -- see (Xian et al. 2018). If the two prior aspects will be addressed by the Authors, I would be more than happy to call for a full acceptance.

*Post-Rebuttal Evaluation [FINAL]*


I have thoroughly inspected the revised manuscript and read the response provided by authors. I was pleased in registering that my two requests were taken into consideration by authors who provided a solid sensitivity analysis and additional experimental evidences on ImageNet. Therefore, taking this into account, I am now convinced in raising my original score (5: Marginally below acceptance threshold) to a full acceptance (7: Good paper, accept).

---

> ### Author Response · Authors · 2020-11-15
> **A New Section to Analyse the Sensitivity of Hyperparameters; More Results on full ImageNet**
>
> Thank you for reviewing our submission and the comprehensive comments! We are very happy to answer the two aspects with regard to sensitivity and datasets!
>
> **Q1**: Sensitivity. The method is depending upon a number of hyper-parameters (such as the threshold for the visual/semantic neighbor graphs or the softmax temperature).
>
> **A1**: We have added one section and Figure 3 to show the effects of the selection for hyperparameters: threshold, temperature, N/K for N-way-K-shot task, propagation steps, etc. For the threshold, we found the performance is similar when the threshold is bigger than cosine(30). This is because when the threshold is too small, it does not allow any edges to connect between classes and thus there is no propagation and will damage the performance. The accuracy float is within 2% when the temperature is chosen between 1-20.
>
> **Q2**: About datasets. Although tiered-ImageNet is a popular benchmark in few-shot learning, there are methods which provide experimental results on the full ImageNet, such as [b]. It is not fully clear to which extent the proposed variants of tiered-ImageNet, adapted for zero-shot learning, would improve upon the existing ImageNet specs.  It would be great to have more experimental pieces of evidence on other benchmark datasets which are usually considered in zero-shot learning.
>
> **A2**: As suggested, we have included the results on ImageNet in Table 5 in the Appendix.
>
> Here is the comparison on ImageNet (hop2)
>
> |Methods|Hit@1(%)|Hit@5(%)|Hit@10(%)|Hit@20(%)|
> |:---|:---|:--|:--|:---|
> |ConSE|8.3|21.8|30.9|41.7|
> |SYNC|10.5|28.6|40.1|52.0|
> |EXEM|12.5|32.3|43.7|55.2|
> |GCNZ|19.8|53.2|65.4|74.6|
> |DGP|26.6|60.3|72.3|81.3|
> |IPN|**27.1**|**61.1**|**73.8**|**82.9**|
>
> Here is the comparison on ImageNet (hop3)
>
> |Methods|Hit@1(%)|Hit@5(%)|Hit@10(%)|Hit@20(%)|
> |:---|:---|:--|:--|:---|
> |ConSE|2.6|7.3|11.1|16.4|
> |SYNC|2.9|9.2|14.2|20.9|
> |EXEM|3.6|10.7|16.1|23.1|
> |GCNZ|4.1|14.2|20.2|27.7|
> |DGP|6.3|19.3|27.7|37.7|
> |IPN|**6.8**|**20.1**|**28.9**|**39.6**|
>
> Compared to full ImageNet with 30k classes, the benchmark (1k classes) we proposed can achieve a better trade-off between the computational cost and benchmarking ability.
>
> [a] Xian et al.  Zero-Shot Learning -- A Comprehensive Evaluation of the Good, the Bad and the Ugly. T-PAMI 2019
>
> [b] Michael et al. Rethinking Knowledge Graph Propagation for Zero-Shot Learning. CVPR 2019.
>
> **If there is any unaddressed concern, please let us know and we are more than happy to discuss.**

---

### Decision · Program_Chairs · 2021-01-07
**Final Decision**

**Decision:**

Accept (Poster)

**Comment:**

Three experts in the field recommend accepting the paper (ratings 7,7,6) after the author response, appreciating the improvements the authors made.
[Note: The AC is mainly disregarding R3's rating, as R3 did not respond to the early request of the AC to clarify their review, did not respond to the authors request for clarification, and did not participate in any discussion past their initial short review.]

The solid experimental evaluation and an original methodology for zero-shot learning speak for accepting the paper.

[The area chair is certain about accepting the paper, but not fully confident if it should be Poster or Spotlight.]